# Pansexuality: A Closer Look at Sexual Orientation

**Arina Pismenny**

Department of Philosophy, University of Florida, Gainesville, FL 32611, USA; arinapismenny@ufl.edu

**Abstract:** 'What is 'sexual orientation' for?' is a question we need to answer when addressing a seemingly more basic one, 'what is sexual orientation?'. The concept of sexual orientation is grounded in the concepts of sex and/or gender since it refers to the sex or gender of the individuals one is sexually attracted to. Typical categories of sexual orientation, such as 'heterosexual', 'homosexual', and 'bisexual', all rely on a sex or gender binary. Yet, it is now common practice to recognize sex and gender categories that transcend the binary. Should our sexual orientation categories be revised to reflect sex and gender diversity? Drawing on the example of pansexuality, I argue that they should. The reason is that one aim of reconstructing the concept of sexual orientation—in addition to the epistemic goal of understanding—should also be political: it should make it easier to argue for the protection of those who have been marginalized or discriminated against because their sexual attraction is other than heterosexual.

**Keywords:** pansexuality; sexual orientation; sex; gender; social justice

## 1. Introduction

What is sexual orientation? One could situate this concept within the biological framework, aiming to shed light on its place in human reproductive and child-rearing practices. One could ask whether it is a natural kind: whether categories such as 'heterosexual', 'homosexual', and 'bisexual' "carve nature at its joints". One could want to clarify the common use of sexual orientation terms in order to aid individuals on dating sites. One could ask whether it is a social construction, given that throughout human history and across cultures, humans have used different concepts to refer to sexual behavior and sexual identities. Such descriptive projects aim to elucidate the concept of sexual orientation and have been pursued by John Corvino [1], William Wilkerson [2], Esa Díaz-León [3], Saray Ayala [4], Kathleen Stock [5], and Raja Halwani [6], to mention but a few.

Alternatively, one can engage in an ameliorative project of conceptual engineering, tailoring the concept of sexual orientation with certain social and political goals in mind. Robin Dembroff develops one such account [7]. Some have proposed to consider certain sexual and romantic dispositions as sexual orientations in order to bring them under the protective umbrella provided by anti-discrimination laws. Thus, Ann Tweedy proposes to treat polyamory as a sexual orientation [8], and Luke Brunning and Natasha McKeever argue that we should recognize asexuality as a sexual orientation in order to promote visibility and protection under the law [9].

In this paper, I use the case of pansexuality [1]—a sexual orientation (as I will argue) that designates attraction not based on sex or gender—to test various conceptions of sexual orientation. My goal is threefold: (1) to explicate the concept of 'pansexuality' as a sexual orientation, (2) to highlight difficulties some accounts face in accounting for pansexuality, and (3) to draw lessons about 'sexual orientation'.

In the next section, I begin by explicating my aims, methods, and theses and outline my argument. Section 3 provides a preliminary sketch of pansexuality by comparing it to other sexual orientations. Section 4 summarizes the standard features of sexual orientation. Section 5 delineates sex, gender, sexual arousal, attraction, and desire and explicates what it means to say that for pansexuals, the sex or gender of their partners does not matter. In



Section 6, I examine three philosophical theories of sexual orientation: Kathleen Stock's Orthodox Account, Robin Dembroff's Bidimensional Dispositionalism, and Raja Halwani's Well-being Condition Account, highlighting the advantages and difficulties each one has in accounting for pansexuality as a sexual orientation. I conclude by emphasizing the need to revise our sexual orientation categories so that, along with descriptive ends, they serve the social and political aims of sex and gender justice.

## 2. Preliminaries: Aims, Methods, Theses, and Argument

I take the importance of sexual orientation categories to be primarily grounded in their various social functions. They describe the diversity of sexual preferences and behaviors and make sense of certain aspects of human sexuality, including one's own. They do so in part by normalizing certain patterns while invalidating others, like the pathologizing of homosexuality in the early 20th century, for instance. Thus, they are both descriptive and normative. They help us navigate our interactions with one another by signaling who is and is not a potential sexual partner. These categories also inform public policy related to spheres such as marriage, employment, and healthcare. Given the socially important functions of sexual orientation categories, I follow Sally Haslanger [10] and Robin Dembroff [7] in their pursuit of engineering categories that best serve our social and political goals. Our mainstream sexual orientation categories—heterosexuality, homosexuality, and bisexuality—are grounded in sex and gender binaries: male/female and man/woman. Yet, as intersex, transgender, and nonbinary individuals gain more visibility, it is clear that neither sex nor gender is binary. Our sexual orientation categories should reflect this fact, not only for the sake of accuracy but also to advance the cause of sex and gender justice. Sex and gender justice includes the following: (1) recognizing the multiplicity of sexes and genders rather than erasing those that do not fit into the sex and gender binaries; (2) protecting vulnerable groups currently marginalized because of their sex or gender from discrimination; (3) empowering individuals to make decisions about their own bodies, sexuality, and gender identity and challenging social norms and expectations that restrict their choices and opportunities; (4) promoting equity by making resources such as jobs, healthcare, and other social goods available in accordance with persons' needs; and (5) weakening oppressive gender norms and stereotypes in order to create a society that is more inclusive, accepting, and affirming of diverse genders and sexes.

Sexual orientation categories have a role to play in this enterprise. By rejecting the sex and gender binaries as their grounding features, the revised sexual orientation categories will affirm the diversity of sexes and genders and facilitate the visibility of sex and gender minorities. In this way, the project pursued here is *ameliorative*. However, I do not propose an ameliorative account of sexual orientation as such. Instead, I use 'pansexuality' to illustrate how treating it as a sexual orientation might help to advance the aims just listed. There are lessons to be learned from 'pansexuality' that could help us reconceive our sexual orientation categories to make them more inclusive of all sexes and genders. As I will argue below, pansexuality's recent popularity attests that when it is viewed as a revision of bisexuality, it promotes the goals of sex and gender justice.

Before we are ready to reap the lessons from pansexuality, we must ask whether pansexuality is indeed a sexual orientation rather than simply a trend that constitutes a sexual identity at best [2]. The answer depends on the theory of sexual orientation at play. I examine three such theories below, concluding that it is a sexual orientation according to some but not others. Since my goals are ameliorative, and since I take 'pansexuality' to be an improved sexual orientation category, I rely on reflective equilibrium to clarify both 'pansexuality' and 'sexual orientation' to arrive at a better, more useful conception of both. I use the commonsense conception of sexual orientation to show that pansexuality has most of the features of sexual orientation. I then use pansexuality to put pressure on our mainstream sexual orientation conception to question one of its core features.

One difficulty with such an approach is that it may seem question-begging. How should we decide on the constraints on a definition of 'sexual orientation'? The three

accounts I consider below appeal to common sense to justify their approaches. Ameliorative accounts are likely to fail to align with our commonsense conception since they aim to be an improvement of that concept. Arguments must be given to defend the selection of the commonsense features we want to preserve so as to avoid cherry-picking.

Thus, a successful ameliorative account of sexual orientation will at least (1) provide a description of sexual orientation that is plausible, coherent, informative, and in line with common sense in a relevant way [5,7], as well as (2) be well-situated to help pursue its ameliorative goals [7].

Similarly, to show that pansexuality is a sexual orientation, it needs to be shown that construing it this way is plausible, intuitive, coherent, and informative. In the rest of the paper, I argue that 'pansexuality' is such a concept. If my arguments are on the right track, then my analysis of pansexuality will contribute to a revision of our sexual orientation categories to make them both more accurate and more inclusive.

I defend two main theses: (1) pansexuality is a sexual orientation, and (2) pansexuality is a good example of how sexual orientation categories can be more inclusive, reflecting the diversity of sexes and genders. To defend the first thesis, I argue in Section 3 through Section 5 that pansexuality possesses the majority of features that characterize a sexual attraction as an orientation. I defend the second thesis by testing three theories of sexual orientation against pansexuality (Section 6). The virtue of 'pansexuality'—its acknowledgment of the existence of multiple sexes and genders—renders some theories more plausible than others. Through this acknowledgment, pansexuality might help advance the aims of sex and gender justice by making the multiplicity of sexes and gender more visible and accepted. I begin with some preliminary remarks on pansexuality.

## 3. Pansexuality: A Prelude

The term 'pansexuality' designates sexual attraction to others regardless of their sex or gender. Recently, pansexuality has been gaining visibility, as many celebrities have come out as pansexual. Jazz Jennings, Youngblud, and Madison Bailey all belong to Generation Z, which suggests that this sexual identity is popular with a certain age demographic [3]. According to a recent Gallop Poll, 21% of Gen Zs (those born between 1997 and 2003) identify as LGBTQIA+ [13]. This is double compared to Millennials (10.5%). There is little data on pansexuals, but according to a 2017 Harris Poll, 2% of LGBTQIA+ people ages 18–34 identify as pansexual. This means that about four Gen Z individuals per thousand identify as pansexual. Of all other age groups, about 1% of LGBTQIA+ identify as pansexual [14].

While this identity may be more prevalent among the younger generations, the term 'pansexuality' has been around for centuries [15]. Bisexuality is a lot more widespread: 57% of LGBTQIA+ individuals identify as bisexual [13], despite having often been invisible and discriminated against by both straight and gay and lesbian communities [16]. On the face of it, 'bisexuality' denotes sexual attraction to members of the same as well as 'opposite' sex or gender, presupposing that there are just two sexes or genders [4]. 'Pansexuality' is more inclusive since it includes individuals of any sex or gender into the potentially sexually desirable pool of partners, including intersex and nonbinary persons.

Some argue that bisexuality should be thought of as an umbrella term that includes pansexuality [17], for some self-identifying bisexuals are attracted to multiple sexes and genders. Given this overlap with 'bisexual', the term 'pansexual' may accurately fit more than 2% of the LGBTQIA+ community [5].

The inclusiveness of the pansexual category may be one reason why it is a popular identity among Gen Zs and Millennials. An Internet search for pansexuality jokes renders two general kinds: the first about kitchenware (e.g., "I recently came out as pansexual. Lately I really like cast iron. I've tried dating Teflon but nothing ever seems to stick") and the second about nondiscrimination (e.g., "As a pansexual I feel equally awkward around everyone"). The latter joke turns on the assumption that pansexuals are attracted to everyone. After all, the prefix 'pan' means 'all' [6]. Yet, this is misleading since pansexuals, like most people, are likely to have particular preferences regarding their partners' physical

appearance and character. Rather, the fact is that for pansexuals, no person is excluded from being a potential object of sexual attraction on the basis of their sex or gender.

Given its inclusiveness, a moralistic pansexual might be tempted to boast that pansexuality is morally superior to other sexual orientations. The fact that the genitals of one's partner do not matter—accepting of penises, vaginas, as well as other variations found in intersex individuals—seems to elevate pansexuality's moral status by avoiding the cissexism of those who refuse to date trans or intersex people. It allows for a wider variety of potential sexual partners, including marginalized persons such as queergender and intersex individuals.

As the notorious failure of conversion therapy attests, however, sexual orientation is not a choice and, therefore, not a candidate for moral superiority [7]. Still, it is important to recognize that pansexuality sets a precedent for more inclusive sexual orientation categories by transcending the sex and gender binaries.

Before I give arguments for pansexuality as a sexual orientation, I describe key features of 'sexual orientation' in the next section and explicate the relationship between sexual orientation, sex, and gender in the following section.

### 4. Sexual Orientation: Some Key Features

Generally, sexual orientation is thought of as a basic stable disposition to feel sexual attraction only towards individuals of a given sex or gender. It differs from sexual preferences that are typically for specific body types or sexual acts. For instance, one might have a preference for full-bodied or slim-bodied sexual partners or for partners with long blond hair. But this preference is usually less basic than one's sexual orientation, less important than the sex or gender of one's partner. As Halwani puts it, "one is homosexual first, and prefers blonds second" [6] (p. 464). Similarly, sex or gender attraction is more basic than the sexual acts that one prefers to engage in. That is, one is a heterosexual first and prefers foot play second.

Sexual orientation is commonly characterized as a *disposition* since it is attributed to individuals as a more or less stable propensity to feel or behave in a certain way given a particular kind of stimulus. Sexual orientation can be thought of as a disposition to experience sexual desire towards individuals of a particular sex or gender or to seek or engage in sexual activity with individuals of a particular sex or gender. Both the desire and the behavior criteria are problematic, however, as sexual desires towards a particular sex or gender are sometimes dismissed as mere fantasies that are not acted upon. It is said that a heterosexual man might have fantasies about having sex with men, for instance. On the other hand, social constraints might prevent one from acting on one's sexual desires—in a society in which homosexuality is criminalized, for example. Conversely, one might find oneself in prison without any available sexual partners matching one's orientation. So, behavior is not necessarily indicative of one's sexual orientation.

One solution proposed by Edward Stein is to conceive of our sexual desires and behaviors under ideal conditions [21]. Were our world free of homophobia, transphobia, and other prejudices and taboos, we would be in a better epistemic position to access our sexual desires and act on them. Then, we would be able to correctly ascribe to ourselves and others the appropriate sexual orientation category. In the real world, however, many people might be psychologically unable to draw realistic conclusions from a thought experiment positing ideal conditions of acceptance [8].

To circumvent these worries, Robin Dembroff proposes to start with the commonsense practice: (1) the operative concept of sexual orientation denotes sexual attraction to persons *on the basis of* their sex or gender; (2) it assumes a reasonable diversity of sexual partners such that circumstantial factors such as being in prison are ruled out; and (3) one is willing and able to engage with others sexually, ruling out possible obstacles to sexual activity such as a medical condition [7] (p. 17).

Esa Díaz-León objects to Dembroff's behavioral characterization of this disposition [22] (p. 299). She argues that we should instead understand it as a disposition to experience

sexual *attraction*, even if one is unable to act on it. A bisexual person committed to a monogamous heterosexual relationship will likely experience sexual attraction to others of the same sex or gender without acting on it. Dembroff might respond that even in such a case, the behavioral condition seems plausible since the bisexual person is able and willing to engage sexually with males and females under normal conditions and might have prior to their marriage. However, many ordinary circumstances can undermine the possibility of acting on attractions that are otherwise stable. For this reason, (3) should be amended to 'one maintains a stable disposition to experience sexual attraction towards others whether or not one is willing or able to engage in sexual behavior to fulfill it' [9].

While this characterization of sexual orientation leaves room for exceptions, I take it that it captures our commonsense practice of ascribing sexual orientation, setting aside circumstantial factors that might distort our sexual desires and behaviors.

We can further agree that sexual orientation is typically ascribed to oneself and to others on the basis of sexual desires and behaviors that track the sex and gender binaries. Homosexuality, heterosexuality, and bisexuality are all grounded in the sex and gender binaries. These sexual orientation categories are *reflexive* since they take the sexes and genders of one's preferred partners to be "the same", "the opposite", or both with respect to one's own [5]. Thus, these categories contain information about the party to whom they are ascribed, as well as about their preference for the sex or gender of their sexual partners.

Sexual orientation is not the same as romantic orientation. The latter refers to a propensity to fall in love with certain kinds of people. The two often overlap: if one is a homosexual, one is likely to also fall in love with people of the same sex or gender. But this need not be the case. Some individuals are asexual but not aromantic. That is, while asexual people may not be interested in any particular kind of person sexually, they might have a preference for a type of person with whom they fall in love. The converse is also true. Some people who are interested in sex and in having a particular kind of sexual partner may be aromantic—unable to fall in love [24]. It is, therefore, important to keep sexual and romantic orientations apart.

In my account so far, I have been referring to the binaries of sex and gender. But which one is it—sex or gender? I will say more about this in the next section, but the commonsense (mainstream) practice of sexual orientation ascription provides no clear answer. Indeed, as Kwame Anthony Appiah argues, a competent language user is not required to rely on the same criteria as more sophisticated speakers to make a successful reference [25]. Many people in the United States do not think that there is a difference between sex and gender. They simply rely on apparent sex when ascribing sexual orientation to themselves or others. On the other hand, a supportive romantic partner of a trans person might change their sexual orientation self-ascription in order to affirm the gender identity of their partner [26]. The former might ascribe 'woman' to a transwoman whose gender presentation aligns with the mainstream feminine presentation, taking themselves to be tracking her sex, while the latter would ascribe the same gender category while knowing that their partner is male. Both will ascribe 'straight' sexual orientation to the transwoman's romantic partner as a result. In short, while competent language users rely on a variety of conditions in their ascription practices, it is clear that sex or gender are the grounding factors of the ascription.

When it comes to the self-ascription of sexual orientation, it might be suggested that we are talking not just about sexual orientation but about *sexual identity*, for sexual identity is constituted in part by self-ascription [27] (p. 120). Yet, one might deny that a man who identifies as straight but has sexual fantasies about other men is actually straight, for instance. He might just be in denial. This would make room for mistaken self-ascriptions of orientation. Matthew Andler defines 'sexual identity' as the social meaning of 'sexual orientation', thus placing 'sexual orientation' as conceptually prior to 'sexual identity' [11].

To sum up, the mainstream conception has these three key aspects: (1) it picks out attraction based on sex or gender; (2) it assumes a variety of potential sexual partners, excluding special circumstances such as being in prison; and (3) it involves a stable disposition to experience sexual attraction towards others, irrespective of the ability or willingness

to engage in sexual activities to fulfill that attraction. So far, I have taken the ordinary ascription of sexual orientation based on sexual and gender binaries mostly at face value. Before I turn to three more searching philosophical accounts of that category, I explicate the concepts of 'sex' and 'gender' and analyze the nature of pansexual attraction.

## 5. Sex, Gender, or Something Else?

Insofar as sexual orientation categories are grounded in sex and gender, we need working accounts of these terms. Although a comprehensive account of sex and gender is beyond the scope of this paper, it is important to differentiate between the two, as assigned sex may not always align with gender, notably in nonbinary, intersex, and transgender individuals. That is the task of the present section and the next. I proceed in part by clarifying the sense in which pansexuals are attracted to others 'regardless of sex and gender'.

### 5.1. Sex and Gender

'Sex' can be defined as a cluster concept that includes gamete size, chromosome type, internal reproductive organs, genitalia, patterns of hormonal secretion and receptivity, and secondary sexual characteristics. Sex categories are traditionally taken to comprise males and females. Anne Fausto-Sterling, however, has argued that we should also include three types of intersex persons, though she holds that even that is a simplification, as the determinants of sex are arranged on a continuum [28,29]. In contrast to sex, 'gender' is defined by social roles governed by a set of rules associated with sex and sexed bodies in social, cultural, and political contexts. Therefore, gender can be construed as the social meaning of sex [10]. Gender rules dictate how individuals act, think, feel, and interact with others.

We usually ascribe gender to others based on their gender presentation and expression [11]. *Gender expression* refers to the manifestations of traits and behaviors that are coded as masculine or feminine. Talia Bettcher argues that *gender presentation* "concerns one's gendered sensory appearance to others." She asks, "Does one look like a woman? Does one sound like a woman? Does one "read" as a woman? While gendered clothing and grooming are obviously relevant, presentation also includes mannerism and comportment" [27] (p. 121). Thus, one can present as a masculine (expression) woman (presentation), as a feminine man, or as an androgynous person, etc.

*Gender identity* is another important concept for this discussion. It refers to one's own inner sense of gender. According to Katherine Jenkins [32,33], gender identity is the experience of gender norms as relevant or irrelevant to oneself. This picture illustrates cisgender and transgender experiences, where individuals' gender identities are experienced as congruent or incongruent with gender norms that are applied to them [12].

One's gender identity likely informs one's gender expression and presentation, as well as contributing to gender ascription by others. Furthermore, gender presentation is often taken to be evidence of one's sex [35]. Clothing, manner, voice pitch, gait, etc., are taken to communicate the genital status of a person. In such inferences, one is relying on the sex binary, ascribing sex on the basis of gender presentation. This is why transgender and intersex people are 'penalized' when their genital status is incorrectly inferred from their gender presentation.

Thus, the relationship between sex and gender is sometimes construed as licensing mutual implication: 'she is a woman, therefore, she is female'; 'he is male, therefore, he is a man'. The inferences rely on and reinforce the sex and gender binaries. In order to acknowledge and respect sex and gender identities that do not fall into the strict cis binary categories, one should, at the very least, hesitate to make such inferences and be open to being wrong about one's inference.

### 5.2. Attraction and Desire

Having sketched out the relationship between sex and gender, we can see how they are implicated in sexual arousal, attraction, and desire.

*Sexual arousal* is a physiological response to some stimuli associated with genital and psychological excitement, such as lubrication and swelling of the sexual organs. It also involves positive mental engagement in response to sexual stimuli [36]. Sexual arousal does not always correspond to one's sexual orientation. It is well-established that women become sexually aroused by a greater variety of stimuli than men [19]. *Sexual attraction* is a disposition to engage with the object of attraction [9]. Sexual orientation is a kind of sexual attraction—one that is typically correlated with some preferred kind of object(s). The common sexual orientation categories discussed in the previous section consist of dispositions to be sexually attracted to and usually sexually aroused by individuals of a certain sex or gender.

*Sexual desire* is a disposition to seek sexual pleasure by engaging with the object of one's desire. The teleology of sexual desire is sexual pleasure [37]. While sexual arousal is the psychophysiological response to some stimuli, sexual desire is an intentional state whose content can be very fine-grained [38]. The content of sexual desires will likely reflect one's sexual preferences with respect to preferred body types of sexual partners, as well as one's sexual orientation in so far as gender features, as well as sex (genitalia), could be points of focus motivating engagement.

### 5.3. Pansexuality As a Sexual Orientation: First Pass

Equipped with these distinctions, I now take a first pass at construing pansexuality as a sexual orientation.

It seems that pansexuality satisfies conditions (2) and (3) of sexual orientation outlined in Section 4. A pansexual is someone who (2) has a reasonable diversity of potential partners and (3) experiences sexual attraction towards others. Condition (1), however, should give us pause since it states that sexual orientation is a stable disposition to experience sexual attraction towards others on the basis of their sex or gender. Pansexuals reject this characterization of their sexual attraction. Instead, they characterize it as 'hearts not parts' [16,39,40]. According to several qualitative studies, pansexuals describe their sexual attraction in the following ways: "When I say I'm pansexual, I mean I am attracted to people of all genders. To me it makes no difference what genitals someone has, or if the ones they do have 'match' the way they present their gender" [16] (p. 181). "[I am] [i]nterested in people over people's bits. Gender is amongst the many variables that literally do not enter the equation of "to date or not to date"" [39] (p.117). "I have the ability to be attracted to any person, wether [sic] they are trans\* or cis or intersex or some other non binary gender/sex. I don't like everyone, but I could" [40] (p. 48) [13]. Pansexuals deny attraction on the basis of sex or gender. This suggests that it is not a sexual orientation since it fails to satisfy (1).

Perhaps (1) could be reinterpreted in the following way. The term 'sexual orientation' is ambiguous. It may refer to sex as an *activity* rather than a feature of one's partner [14]. Clearly, pansexuals are oriented towards sex in this sense. If that is sufficient for X to be a sexual orientation, then pansexuality is indeed one.

An analogy with asexuality is helpful here. Brunning and McKeever argue for construing asexuality as a sexual orientation for the sake of promoting visibility and affording social and political benefits to asexuals as a protected class. They demonstrate that asexuals may experience sexual desires that can be disconnected from attraction to specific individuals and instead be directed at situations, roles, or scenarios. [9] (p. 504). Thus, we can move away from construing sexual orientation categories as being necessarily grounded in sex or gender [9] (p. 501). Pansexuality may be similarly understood without reference to the sexes or genders of potential partners.

Some may object that pansexuals are mistaken or insincere when they describe their sexual attraction. Kayley Vernallis, in her analysis of bisexuality, argues that attraction that is not based on the sex or gender of one's sexual partner is nonsensical [42]. She does not differentiate between sex and gender, using 'gender' to refer to both. She also takes it that there are only two genders—man and woman.

To make her point, Vernallis distinguishes between two kinds of bisexuals. Gender-specific bisexuals are those who experience sexual attraction to others on the basis of their gender. Gender-nonspecific are those attracted to others regardless of their gender. Gender-specific bisexuals are attracted to "women as women, with their range of bodily traits (breasts, vagina, capacity to bear children, larger hip-to-waist ratio, etc.) and, often, with the character traits (warmth, empathy, etc.) associated with women" [42] (p. 164). Similarly, they are attracted to "men as men, with the range of bodily traits (penis, facial and chest hair, physical strength, larger shoulder-to-waist ratio, etc.) and, often, the character traits (competitiveness, assertiveness, etc.) associated with men" (ibid.) [15]. Vernallis argues that a bisexual woman will have qualitatively different sexual experiences with women and men because she can feel more dominant or masculine having sex with a woman and more submissive and feminine having sex with men.

Gender-nonspecific bisexuals, on the other hand, view gender as irrelevant to their sexual attraction. Instead, they might be said to be attracted to the person *themself*, i.e., their character and personality [16]. Vernallis denies that such an attraction is possible because societal expectations about character and gender "make our sexual attraction more gender-laden than we think" [42] (p. 166). Thus, even if a person says that they are attracted to the other's character and personality, character and personality are gendered, making our attraction gendered. Vernallis thus rejects gender-nonspecific bisexuals as a possible sexual orientation.

A similar distinction can be applied to pansexuals. There may be gender-specific and sex-specific pansexuals. There may also be gender-nonspecific and sex-nonspecific pansexuals. Since pansexuality is more inclusive than bisexuality, gender-specific pansexuals are attracted to their sexual partners on the basis of gender, of which there are more than two. Sex-specific pansexuals are attracted to their sexual partners on the basis of their sex, of which there are also more than two. We can make sense of this by saying that genitals, secondary sexual characteristics, as well as gender features such as manner of dress, speech, assertiveness, etc., can all figure in the intentional content of one's sexual desire.

Can we make sense of gender- or sex-nonspecific pansexuals? When pansexuals say they are attracted to others regardless of their sex or gender, they mean that no one is excluded from the pool of potential sexual partners on the basis of sex or gender. To that extent, sex and gender are irrelevant. Nevertheless, an important distinction can be made. To be sure, gender identity, expression, and presentation all speak to the person's self. But pansexuals may be drawn to others on the basis of their vitality, charisma, charm, and grace. Sophia Loren expresses this banality by saying, "sexiness comes from within . . . it really doesn't have much to do with breasts or thighs or the pout of your lips." Pansexuals may be responding to the sex appeal that is not grounded in sex or gender qualities [17]. Instead, they are responding to the ways in which a person is intelligent, confident, empathic, or driven. While these qualities are inflected by gender norms and identity, they may also transcend them [18].

Vernallis worries that in such attraction, bodies do not matter. But this is not so. Since individuals are embodied, their selves (personality traits, behavior, and attitudes) are manifested through their bodies—the ways they present and act in the world. Since in sex, we typically engage with others' bodies, even sex- and gender-nonspecific pansexuals will need to figure out what their partners find stimulating and enjoyable given their anatomy and gender identity. Even for those who have no preference, since genitals tend to be erogenous zones, they will likely be the focus of stimulation. Bodies do matter [19].

That pansexuals do not exclude anyone on the basis of sex and gender does not mean, however, that pansexuals do not enjoy the sex and gender of their partners. Rather, we can expect a wide range of preferences among pansexuals as to the gender expression and presentation of their sexual partners. While some pansexuals may attend to and primarily delight in the nongendered qualities of their partners, others may simply not perceive those qualities as gendered or sexed.

I conclude that pansexuality might be a sexual orientation even if we reject the presupposition that this requires us to characterize sexual attraction on the basis of sex or gender.

That highlights the fact that whether pansexuality is a sexual orientation depends in part on what we take a sexual orientation to be. Hence, I now turn to an examination of three philosophical accounts of sexual orientation.

## 6. Sexual Orientation: Three Philosophical Accounts

The three views discussed in this section are the *Orthodox Account* defended by Kathleen Stock, Robin Dembroff's *Bidimensional Dispositionalism*, and the *Well-being Account* advocated by Raja Halwani. The three accounts differ in what they aim to do. As a result, they differ in their answers to the question of whether sexual orientation is or should be grounded in sex or gender (condition (1)). Applying each view to pansexuality will expose some weaknesses in each account, leading us to reflect on what sexual orientation categories do and should do for us.

### 6.1. The Orthodox Account

According to what Kathleen Stock dubs the Orthodox Account [OA], sexual orientation is sexual attraction towards others on the basis of sex, not gender [5]. Stock argues that there are primarily two sexes: male and female, thus affirming the sex binary. Each of these is a cluster concept, not one defined by some essential conditions: "Sex is appropriately characterized in terms of a cluster of endogenously produced morphological, genetic and hormonal features" [5] (p. 300). None of these features is essential. Yet, possessing some combination of these characteristics is sufficient to produce the male or the female sex. The two sexes are biologically determined rather than socially constructed [45]. Even if 'sex' is a cluster concept, it is clear that for Stock, its most relevant features are the genitalia and secondary sexual characteristics (pp. 301–302). It is not the gamete size, the chromosomes, or the hormones that figure in one's attraction.

How does this view deal with intersex identity? Stock argues that cases of intersex people should be viewed as anomalous deviations from normal development. Since she takes 'sex' to be a cluster concept, she argues that individuals whose sex is truly indeterminate are a lot rarer than Anne Fausto-Sterling [28,29] states: "The vast majority of the cases subsumed under Fausto-Sterling's 1.7% turn out to be easy to identify as on one distinctive gamete producing-pathway or other . . . For instance, Non-classical Congenital Adrenal Hyperplasia (responsible for 1.5% of the 1.7% in Fausto-Sterling's calculation) is compatible in females with pregnancy and carrying a baby to term" [45] (p. 30). Thus, Stock reasons that the real number of intersex individuals is about 0.02%. It seems that Stock thinks that because this number is so small, it does not warrant recognizing 'intersex' as a separate sex category or multiple categories. This preserves the sex binary and allows Stock to endorse the dictionaries' definition of 'woman' as 'adult human female' and 'man' as 'adult human male' [45,46]. This definition seems compatible with the idea that 'gender' encapsulates social meanings of sex. But it clearly entails that the term 'woman' correctly applies only to females.

Stock's view that only sex can ground 'sexual orientation' is based on the claim that there exists no adequate account of 'gender' that could do so. Her reasons for this are that the sex of the desired object is often part of the intentional content of one's sexual desire; that sex inflects our interpretation of masculine and feminine features; and that if sex and gender are both social constructions, then there is no real difference between them [20]. She also argues that sexual orientation's grounding in sex provides the best explanation for the fact that [some] lesbians refuse to date transwomen.

The OA thus follows what I called the 'commonsense' account in recognizing only three sexual orientations: homosexuality, heterosexuality, and bisexuality, although Stock understands bisexuality as a "compound disposition" comprised of homosexual and heterosexual dispositions [5] (p. 299). The OA also takes sexual orientation to be reflexive. As Stock recognizes, her account is incompatible with pansexuality because pansexuals are not attracted to their sexual partners on the basis of their sex [21]. Nor is 'pansexuality' a

reflexive category. She suggests that pansexuality might be regarded as an identity, not an orientation [5] (p. 308).

To sum up, according to the OA, sexual orientation is sexual attraction towards others on the basis of their biological sex, of which there are only two. The account is informative because it defines orientation as reflexive, as well as in its claim to explain why sex rather than gender is the basis for the disposition to experience sexual attraction towards others.

The OA is defective in several ways. Here, I note just three.

First, Stock's characterization of 'sex' as a cluster concept seems insufficient to justify explaining away the existence of intersex persons. Perhaps she is thinking of 'sex' as a natural kind in which the clustering of properties results from a homeostatic mechanism [47]. A homeostatic property cluster (HPC) can be stable when a mechanism ensures that deviations from the cluster remain rare. Evolution, for example, may preserve a certain kind of "normality" by favoring some types over others, perhaps by lowering the deviant type's rate of reproduction. But people with intersex conditions can often reproduce [48]. And when they cannot, that is often due to the forced surgeries they receive as infants [22]. The plausibility of a cluster concept view of sex actually draws attention to the multiplicity of criteria on the basis of which sex is ascribed, and their many ways of combining are best viewed as giving rise to multiple sexes [23] A phenomenon's rarity does not make it any less real.

Second, given Stock's characterization of sexual orientation as a preference for particular genitals together with secondary sexual characteristics, some individuals may be sexually oriented towards intersex people. One could, of course, insist that preferences for intersex people should be disparaged as fetishes rather than recognized as sexual orientations. But a bias in favor of the statistically frequent has not, in general, been a badge of enlightenment. The intersex identity has been gaining visibility thanks to activists who have been fighting, among other things, against forced 'corrective' surgeries performed on intersex babies with ambiguous genitalia. Given that such orientations are possible, it is reasonable to hope that intersex individuals will become more recognized, respected, and empowered [24]. A humane policy would recognize more than two sexes, and our sexual orientation categories should reflect that [25].

A third reason why OA is defective is that in sexual attraction, gender does matter. One is not attracted to abstract biological types or even to genitals but rather to the individuals with observable gender presentation and expression. Even though Stock insists that a person's masculinity and femininity are inflected with our prior understandings of their sex, this is beside the point, for gender features can be part of the intentional content of one's sexual desire. One might be attracted to how a person 'does' their gender in light of their gender identity, social norms, as well as their sex [26]. In addition, as I have already argued, just because certain clusters of gender characteristics are often interpreted as proxies for sex, this need not be the case. Given that more people identify as trans and nonbinary, as well as the fact that some people are intersex, gender expression and presentation vary drastically. This makes inferences from gender presentation to sex increasingly moot.

These reasons also provide a rebuttal to Stock's claim that pansexuality is only an identity and not also an orientation [27]. On her view, pansexual identity may be had by bisexuals, who are attracted to all sexes, of which there are only two. I have been arguing that there are more than two sexes and that gender also matters for sexual orientation. Pansexuality is not simply an identity that some bisexuals can have.

The OA should be put to rest. It does not capture our intuitions about sexual attraction and is out of step with the growing visibility and acceptance of sexes and genders beyond the binary. Its use impairs the pursuit of sex and gender justice.

*6.2. Bidimensional Dispositionalism*

Robin Dembroff defends a very different account of sexual orientation. Unlike Stock, who aims to clarify and defend the traditional conception, Dembroff proposes a revisionary account. They argue that an improved conception will facilitate legal and social protections for queer identities such as transgender, gender-nonconforming, and intersex persons [7] (p. 7).

Dembroff argues that sex and gender are different but that both must be included in conceptualizing sexual orientation. That is the aim of their proposed "Bidimensional Dispositionalism" [BD]:

A person S's sexual orientation is grounded in S's dispositions to engage in sexual behaviors under the ordinary condition[s] for these dispositions, and which sexual orientation S has is grounded in what sex[es] and gender[s] of persons S is disposed to sexually engage under these conditions [7] (p. 18).

Dembroff's two dimensions allow us to recognize that one's attraction can be grounded in either sex or gender, or both. In addition, BD has several striking features. First, it stresses that sex and gender pick out different features of one's preferred sex partners. It thus recognizes that sex need not align with gender in a cisnormative way [28]. Second, it transcends both sex and gender binaries, as it recognizes multiple sexes and genders, each of which can be a basis for one's sexual orientation. Third, BD is non-reflexive because it identifies orientation by specifying the sexes and genders of one's potential sexual partners rather than as the "same" or "opposite" to one's own. Katie, a cis woman, and Frank, a cis man, have the same sexual orientation if they are both attracted to women (gynosexual) or if they are both attracted to males (androsexual) [29]. The distinction between sex and gender in our sexual orientation categories can account for individuals who are sexually oriented towards trans or nonbinary people, breaking away from the binaries. It makes the resulting categories of sexual orientation more informative as well as more inclusive.

Dembroff also notes that sex and gender are not the only factors that act as filters for sorting potential partners as more or less desirable. "Druthers" are features of one's preferred sexual partners that are not sexed or gendered. Someone might be attracted to tall people or blond people, regardless of their sex or gender [30]. Dembroff chooses not to view such cases as constituting a sexual orientation because that would contradict the everyday practice of relating orientation only to sexes and genders. The main reason that Dembroff keeps sex and gender and druthers apart is that "No one is interested in creating nondiscrimination laws to protect people attracted to blondes or baritones. We are, though, interested in creating legal and social protections for queer, transgender, gendernonconforming, and intersex persons" because they are targets of discrimination or ill-treatment [7] (p. 7). This is why Dembroff rejects druthers as grounds of sexual orientation.

What can BD tell us about pansexuality? If pansexuals are not attracted to others on the basis of either sex or gender, it seems that BD cannot characterize it as a sexual orientation. Yet, as Dembroff concedes, the distinction between sex, gender, and druthers is somewhat arbitrary. That may be a reason to weaken condition (1) to include druthers, which might allow us to classify pansexuality as an orientation. A pansexual might have a preference for a druther, such as height or body shape, that acts as a filter comparable in strength to orientations grounded in sex and gender. If all pansexuals were to have a type, then 'pansexuality' could be eliminated and replaced by categories that clearly reflect these types. 'Long-hair-sexual', 'blue-eye-sexual', 'full-body-sexual', etc. However, there is no reason to assume that all pansexuals have types and that they would rather be identified by their type. Furthermore, pansexual identity is important to many pansexuals. Pansexual Pride Day is often a time of advocacy and awareness-raising about this identity that is so often rendered invisible. An ameliorative account of sexual orientation should be able to define it in such a way as to make sense of this identity.

In fact, BD does seem to be compatible with pansexuality, after all. To show this, Dembroff might argue that the term 'pansexual' implies not excluding anyone on the basis of sex or gender from the potential sexual partner pool and is, therefore, compatible with BD. Pansexuality captures both dimensions of BD since pansexuals are disposed to engage with any sex and gender. In this sense, pansexuals have a sex and a gender disposition.

Summing up, Dembroff's BD is a revisionary account aimed at the political project of sex and gender justice. To further this objective, they propose that we differentiate between sex and gender when applying sexual orientation categories. BD is a significant improve-

ment on AO because it is intuitive and inclusive. It is well-equipped to accommodate a variety of sexes and genders and is compatible with pansexuality.

At the same time, Dembroff's commitment to condition (1) might be regarded as too conservative. The example of pansexuality suggests that there might be a good reason to relax the distinction between sex and gender and druthers. This brings us to Raja Halwani's account, which goes beyond sex and gender in defining sexual orientation.

### 6.3. The Well-Being Condition Account

Raja Halwani's descriptive account of sexual orientation privileges its connection to well-being. It, too, is revisionary, but unlike the previous accounts, it allows for an orientation that is not defined in terms of sex or gender, thus dispensing with condition (1). Halwani's aim is not to come up with an ameliorative account, though he recognizes that it might turn out to be one anyway [6] (p. 467n8).

Halwani adopts the term sex/gender to capture the most common basis for sexual orientation. It encompasses genitals, secondary sexual characteristics, gender expression and presentation, as well as gender identity. The term captures the non-binarity of both 'sex' and 'gender'. His account appears to be neutral with respect to the reflexivity of sexual orientation. If it is open to non-reflexivity, there is potentially no limit to the variety of such categories.

On this view, the well-being condition is a necessary one for sexual orientation. "Sexual disposition is an orientation [only] if the inability to act on it renders its possessor's life sexually deprived" [6] (p. 469). Sexual deprivation is the deprivation of the profound sexual pleasure that can be experienced in sexual activity. This kind of pleasure is not reducible to physical sensations. One who is deprived of profound sexual pleasure can nonetheless enjoy sex and derive satisfaction from it. But such a person would be missing something crucial—the pleasure of connecting sexually with someone for whom they have a deep preference. To borrow Halwani's example, consider Iman, a lesbian cis woman who is unable to have sex with women because she is married to a man due to the heteronormative pressures in her culture. Iman can still enjoy sex with her husband, but her life lacks the deep and exciting sexual pleasure she can only experience by having sex with other women.

Profound sexual pleasure is an objective human good. For allosexuals (those who experience sexual attraction to others), the inability to pursue such pleasure is a kind of loss that takes away from a life well-lived. The well-being condition stresses that sexual orientation is important to us because our well-being is significantly impacted by it. The satisfaction of that condition may depend on criteria that are not limited to sex or gender. This is the revisionary side of Halwani's account. If such deprivation is not due to one's sexual partners' sex/gender, it might instead come from other criteria. These might relate to particular kinds of sexual acts (being anally penetrated) or to a preference for partners of a certain age group (chronophilia) or body type. Hence, Halwani's account embraces the possibility of defining a given sexual orientation in terms of what Dembroff characterizes as mere druthers. It allows for a limitless variety of sexual orientations, no matter how narrow or uncommon. It is also compatible with Dembroff's BD in cases where sex and gender are a part of the well-being condition.

Pansexuality provides an interesting test case for Halwani's account. If condition (1) does not apply, it might seem that no sexual encounter will leave a pansexual significantly deprived just because their partner is of the 'wrong' sex or gender. If, for pansexuals, the sex and gender of their partners do not matter, engaging with anyone regardless of sex or gender could lead to sexual satisfaction even in a homophobic society that forbids them certain partners. How, then, is the well-being condition to be applied to pansexuals? One possibility is that some features usually regarded as druthers are essential to some individual pansexual's well-being. However, this move would lead us to dissolve pansexuality into orientations that are demarcated by a particular feature on the basis of which a given pansexual person is attracted to others. I have already considered and rejected this move in my discussion of applying BD to pansexuality.

Still, a pansexual may experience frustration and severe deprivation if they are unable to experience sex with individuals of a wide variety of types. That variety of partners is not constrained by sex or gender, but it may need to include many sexes and genders. That is, it could be important for a pansexual individual that there exist a pool of potential sexual partners of diverse sexes and genders. If this is right, the well-being condition is violated for a pansexual if they find themself unable to engage with people of different sexes and genders. Such a person would be gravely sexually deprived.

An objector might point out that there is no reason to think that pansexuality entails preferring sexual diversity of actual as opposed to potential partners, for a pansexual might also prefer monogamy. By the same token, someone might prefer having multiple sexual partners regardless of their orientation.

This seems right. For this reason, it is the pool of *potential* partners that is relevant to the well-being condition. Just as Iman was deprived of the sex essential to her well-being, so a pansexual would be severely sexually deprived if, due to unjust laws, social conventions, etc., they could not act on their attraction [31]. Therefore, a pansexual's experience can fail to satisfy the well-being condition, proving that pansexuality, after all, is a sexual orientation on Halwani's view.

Each of the three philosophical accounts just discussed aims to demonstrate its plausibility not only through logical consistency and compatibility with empirical evidence but also by attempting to show its intuitiveness through highlighting features that are in line with the mainstream understanding of 'sexual orientation'. I have argued (Section 5.3.) that pansexuality is a sexual orientation even if condition (1) (requiring sexual orientation categories to be grounded in the sex or gender of the desired object) is not held to be necessary. I proposed that 'sexual orientation' may be understood as being oriented towards sex as an activity rather than a feature of one's partner. In this section, I have given reasons for rejecting OA because of its commitment to the binarity of sex and its rejection or relevance of gender. Those features render it incompatible with pansexuality. I have argued that pansexuality can be understood as a sexual orientation under BD, which is committed to condition (1), by interpreting (1) as pansexuality's acceptance of all sexes and genders on the one hand and its rejection of their relevance on the other. Admittedly, for sex- and gender-nonspecific pansexuals, this interpretation is tentative since they reject sex and gender as grounding their attraction. I have also argued that pansexuality is compatible with the Well-being Condition Account by suggesting that such a condition is, in principle, specifiable for pansexuals by analogy with homosexuality, where pansexuals would be sexually deprived if they were unable to engage sexually with those to whom they are attracted. Unlike OA, both BD and the Well-being Accounts are better suited for pursuing the ameliorative goals outlined in Section 2. I conclude by returning to the main question—what is sexual orientation for?

## 7. Conclusions

The three philosophical accounts represent different approaches to answering that last question about the point of sexual orientation. Stock's and Halwani's accounts are descriptive, Halwani's and Dembroff's accounts are revisionary, and Dembroff's account is also ameliorative. Each one starts from a different point and arrives at a distinct result. Stock defends the commonsense cisnormative view, which leads her to affirm the sex binary and reject gender. Dembroff distinguishes between sex and gender, proposing to understand sexual orientation as being comprised of these two dimensions. Halwani explicates a core feature of sexual orientation—its significance to well-being. In every case, some features of potential sexual partners, sometimes but not always sex or gender, are defining of sexual orientation. For Halwani, a lack of such a feature in one's sexual partner may constitute sexual deprivation. Dembroff and Stock both rely on sex (and gender) to keep their accounts in line with the commonsense conception. Halwani expands the list of possible grounding features of 'sexual orientation' to any feature that satisfies the well-being condition. While Stock aims at preserving the three mainstream sexual

orientation categories, Dembroff's and (possibly) Halwani's non-reflexive accounts call for some relabeling of sexual orientation categories.

The features of each account are illuminated by applying it to pansexuality. On Stock's view, pansexuality is not a distinct sexual orientation. It is reducible to bisexuality, which is one of just three sexual orientation categories. On Dembroff's account, pansexuality is a sexual orientation because pansexuals are attracted to people of all sexes and genders, even though for some pansexuals, sex and gender may not be the basis of their sexual attraction. Similarly, on Halwani's account, pansexuality is a sexual orientation since the well-being condition is specifiable for it even though it does not necessarily rely on potential partners' sex or gender.

'Pansexuality' is a conventional yet inclusive category. It is conventional in that, following the commonsense practice, it refers to sex and gender, even though it rejects them as grounds for exclusion from the class of potential partners. It is inclusive because it embraces the plurality of sexes and genders. Pansexuality itself appears as a revisionary expansion of bisexuality.

Since I take sexual orientation to be a concept primarily of social significance, my answer to the question 'what do we need 'sexual orientation' for?' has been that in addition to providing an informative explanation for our basic dispositions for sexual attraction and behavior, it is also needed for legal protections, distribution of social goods such as healthcare, and facilitating conditions for inclusion and visibility of sexual and gender minorities. Which kind of account best serves our aims? What kinds of categories do we need to make our lives better? The recognition of pansexuality as an orientation can play two roles in answering these questions. On the one hand, the variety of its manifestations makes the point emphasized by Dembroff that we need 'sexual orientation' to promote sex and gender justice. Its acceptance as a sexual orientation signals the rejection of cisheteronormativity and makes our categories more inclusive for the marginalized individuals of queer sex and gender identities. On the other hand, since pansexuality rejects sex and gender as criteria of exclusion, it welcomes Halwani's well-being condition, taking sexual orientation beyond sex and gender. It decentralizes them as foci of sexual attraction and opens up the possibility of restructuring sexual identities and social norms in ways that are less rigidly constrained by binary conceptions of sex and gender.

The recognition of pansexuality as a sexual orientation is a way to celebrate sex and gender diversity since it is an inclusive category in two senses. First, it literally includes everyone as potential sexual partners. Second, it is likely applicable to many self-identifying bisexuals. In cases where 'pansexual' applies, it should be used instead of 'bisexual' because it helps break away from oppressive binaries. In this way, pansexuality can play an instrumental role in the pursuit of sex and gender justice goals (1), (3), and (5) described in Section 2 by making nonbinary and trans identities more visible and normalized. Pansexuals like bisexuals and homosexuals are likely to face discrimination because they deviate from the heterosexual standard. Because of this, they, too, require legal protections and access to social goods outlined in sex and gender justice goals (2) and (4). At the moment, it is clear that the prolific use of this category by Gen Zs reflects that generation's social and political awareness, as well as self-awareness, and thus attests to the hermeneutical benefit of its availability [32].

**Funding:** No funding was received for this project.

**Institutional Review Board Statement:** Not applicable.

**Informed Consent Statement:** Not applicable.

**Data Availability Statement:** Not applicable.

**Acknowledgments:** A very special thanks to Raja Halwani and to three Anonymous Reviewers. Their insightful comments were tremendously helpful in writing this paper.

**Conflicts of Interest:** The author declares no conflict of interest.

## Notes

1. Another term used to designate such an attraction is 'omnisexuality'. The term 'panromantic' describes a romantic orientation of people who can fall in love with persons of any sex or gender. In this paper, I concentrate on sexual rather than romantic orientation.

2. For an in-depth discussion of 'sexual identity', see [11,12].

3. Notable pansexuals among Millennials (1981–1996) include Miley Cyrus, Rina Sawayama, Brendon Urie, Janelle Monáe, Caldwell Tidicue, better known as Bob the Drag Queen, and Emily Hampshire, among others.

4. I do not mean to imply that sex and gender are interchangeable. I will have more to say below about the distinction and how the two categories figure in our conceptions of sexual orientation.

5. Though, it is also possible that among bisexuals, some do not experience attraction to noncis people.

6. Indeed, the etymology might suggest that pansexuals' sexual attraction expands beyond human adults, incorporating children, nonhuman animals, and inanimate objects. Most pansexuals interpret their own orientation as relating only to adult humans. I shall not here consider the more literal interpretation of the 'pan' in 'pansexuality' as it does not bear on my main argument. For satire on this theme, see Woody Allen's "Sorry, No Pets Allowed" from *Zero Gravity* [18].

7. See Halwani [15]. Halwani holds a nuanced position regarding sexual orie that is not captured by the simple but popular 'born this way' slogan. For discussion, see also Lisa Diamond [19]. In earlier debates about the morality of homosexuality, one position taken by religious and conservative pundits was that while homosexuality in itself is not immoral, acting on one's homosexual desires is. This might suggest that while any given sexual orientation is amoral, behaving in accordance with one's sexual desires can be immoral. That is notably the position recently expressed by Pope Francis [20]. We might agree that the distinction between amoral orientation and immoral behavior might be applicable to some sexual attractions, such as pedophilia.

8. For additional criticism, see Robin Dembroff [7].

9. For an in-depth analysis of Debroff's and Díaz-León's methodological approaches, see Matthew Andler [23].

10. Judith Butler [30,31] has argued that 'sex' itself is socially constructed and so has a social meaning too. As a result, some have argued that the distinction between sex and gender is not useful. I am not committing myself to any particular ontological position regarding either category. Regardless of whether or not both sex and gender are socially constructed, the distinction between them is useful because the distinction helps to make sense of the experiences of transgender, nonbinary, and intersex individuals.

11. Although it is said that we are assigned sex at birth on the basis of the external genitalia, it can be said we are assigned gender at birth since it is decided on the basis of our genitals whether we are raised as boys or girls to become men and women, respectively.

12. For an insightful discussion of genderqueer identities, see Robin Dembroff [34].

13. Of course, self-report and surveys as research tools suffer from many limitations. More work is needed to better understand pansexual sexual attraction that utilizes other methods.

14. I am not saying that pansexuals are oriented towards some particular sexual activities, such as golden showers, or positions, such as missionary. I mean simply that pansexuals are interested in having sex. For a discussion of what makes an activity sexual, see Halwani [41].

15. Given the diversity of gender norms, these examples seem remarkably subservient to narrow stereotypes about gender and sex appeal.

16. For empirical evidence of gender-nonspecific attraction in bisexuals, see Diamond [43], especially Chapter 6.

17. I do not mean to deny that physical appearance makes no difference. But it is clear that numerous individuals considered physically unattractive are nonetheless found to be sexy. Socrates is one case in point.

18. A study on bisexuals and pansexuals found that (1) pansexuals were less likely to use binary and gendered language when describing their attraction, and (2) pansexuals were less likely to describe their attraction to sexes and genders in terms of degrees of strength compared to bisexuals [39] (p. 120).

19. Granted, sexual pleasure can sometimes be achieved entirely through mental stimulation regardless of one's sexual orientation [44].

20. Stock considers three different arguments for thinking that gender rather than sex is what grounds sexual orientation. She rejects all three in favor of sex. For the lack of space, I only briefly sketch her arguments. For more detail, see [5] (pp. 303–305).

21. Stock also says that 'asexuality' is not a sexual orientation but "the absence of one" because there is also a lack of sexual attraction to individuals on the basis of their sex [5] (p. 300).

22. One might insist that in such cases, an intersex person should be classified as either male or female since they must possess gametes of a certain size to reproduce. However, sexual orientation is not defined by attraction to gametes. This is why for Stock, it is the genitals and secondary sexual characteristics that matter more.

23. As Lisa Diamond aptly puts it, " . . . [T]he only clarity and consistency in matters of sex and gender comes from *culture*, and not biology, because it is culture that tells us which body parts matter (for categorization, say) and which body parts do not" [19] (p. 85; emphasis in the original).

24. Dembroff entertains the possibility of trans-oriented people "who experience strong or exclusive sexual attraction to transgender persons" [7] (p. 11).

25　Again, thanks to the work of the activists for intersex rights, the reduction or complete elimination of forced 'corrective' surgeries on intersex babies will provide intersex individuals with an opportunity to decide how and whether to alter their bodies, as well as choose the sex category they identify with which could well be male or female instead of intersex.

26　According to one study, 8.2% of interviewed cis gay men were open to dating trans men, which suggests that for some gay men, attraction for gender and sex comes apart [49]. For a first-person account of a gay trans man in gay spaces, see [50].

27　I acknowledge that pansexuality is also an identity. For the lack of space, I do not here develop a positive account of pansexuality as a sexual identity. For discussion, see [16,39,40].

28　Dembroff explains that while both sex and gender may be at least partly socially constructed, they are nonetheless different categories as they latch onto different physical and social features and serve different purposes [7] (p. 9).

29　Terms like 'androsexual' or 'gynosexual' are ambiguous since it is not clear from their use whether they refer to the sex or the gender of the person.

30　It is true that some druthers might constitute one's gender expression or presentation. For instance, long hair in our culture is typically viewed as feminine.

31　I am, of course, assuming that pansexuals' sexual engagement is constrained by ethical concerns such as the necessity of consent, etc.

32　I thank Anonymous Reviewer 3 for highlighting this point. For data on Gen Z and their values, see [51].

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
