# Peer review of "Pansexuality: A Closer Look at Sexual Orientation"

_philosophies, doi:10.3390/philosophies8040060_

Round 1
Reviewer 1 Report
The essay makes some interesting observations and is generally well written overall. The topic will be of interest to readers. However, it fails to meet its promise of demonstrating what categories of sexual orientation do, or should do, for us. Thus, it fails to make a compelling argument.
Much more focus is needed throughout in detailing the analysis of the positions of Stock, Dembroff, and Halwani. They are each dealt with too lightly to demonstrate convincingly either the significance of their own positions or their significance for the author’s position. This could be helped by more attention to development of the logical structure connecting the key claims to a central argument or position posed by the author, Instead, the author simply posing a broad (and ultimately unanswered) question about what categories of sexual orientation have to offer, or not.
The essay does a decent job of setting out the different categories of sexual orientation, and noting the distinction between sex and gender. However, more clarity is needed at several places, for example, with regard to the difference between expression and presentation, and what is meant by “manifestations of traits”. More importantly however, is the concept of ‘sexual’ itself. It is unclear why pansexuality is a category of sexual orientation if it is not necessarily oriented to sex or gender traits at all. Is it about some other kind of intimacy? This may be a more promising avenue for the author to explore.
At various places the author separates pansexuality from sex and gender, but at other places concedes that our perspectives are “inflected” with gender. The author needs to unpack the concept of ‘sexual’ if it is to do the work that they seem to want it to do. It would also be useful to unpack what they mean by ‘inflection’, for example, how far do gender norms penetrate and colour our perceptions and understandings of others?
I appreciate that pansexuality excludes nobody, but in that case, I had difficulty grasping what makes it an orientation as such? It seemed to me to be more like an absence of orientation – something that exists outside of the defining limits that characterise other ways of regarding people.
At various places throughout, claims are made without sufficient clarity around how they constitute part of an overall argument, rather than rhetorical pronouncements, eg the reference to Sophia Loren (line 314) and lines 421-424.
Author Response
Dear Reviewer 1,
Please see my responses to the comments attached.
Thanks!

Reviewer 2 Report
Generally, I find the paper to be illuminating, well-structured, and – as far as I can discern – to be rooted in relevant literature. I find it ready for publication, and think it will play a constructive role in a special issue on love and sex. I shall therefore limit myself to only a few comments that may improve it further.
1) I think the introduction could do a somewhat better job to describe the general aim and structure of the paper. On the same note, a bit more metatext (“why do I engage with X,Y,Z in the following” or “Why have I….in the above” may make the text more accessible.
2) While the practical, medical and moral implications of accepting or rejecting the various definitions may be clear to those very familiar with the debate, it may be helpful for many readers to take stock of these implications once in a while (e.g., in section 5).
3) Engaging with conceptual engineering. This is certainly relevant vis-à-vis Dembroff’s account, which, superficially at least, shares features of Haslanger’s highly revisionary account of gender (and the accompanying criticisms.) not to put too fine a point to it: should our concepts (regarding things like gender, sex, sexual orientation etc.) reflect some policy/moral-independent reality, or should they be tailored – conceptually engineered – to fit some normative agenda?
Minor points:
P.9:“There are at least two reasons to deny that there are only two sexes. First, a phenomenon’s rarity doesn’t make it any less real. Given the multiplicity of criteria on the basis of which sex is ascribed, their many ways of combining are best viewed as giving rise to multiple sexes.2 -> I think this should be unpacked.
Note 2: “Notable pansexuals among Millennials (1981 – 1996) include Janelle Monáe, Caldwell Tidicue, better known as Bob the Drag Queen, and Emily Hampshire among others.” -> maybe due to not being a millennial, I have never heard of any of these
Author Response
Dear Reviewer 2,
Please see my responses to the comments attached.
Thanks!

Reviewer 3 Report
The author is pursuing an interesting topic and I think there is a lot of potential in the paper. I do think that there are some structural concerns that need to be addressed before it would be ready for publication. The author (p. 1) states that there are three main goals: (1) explicate pansexuality as both a sexual orientation and identity; (2) highlight how current descriptive and ameliorative accounts of sexual orientation struggle to adequately include pansexuality; and (3) draw some general lessons about sexual orientation from the discussion in (1) and (2). I think each of these goals partially succeed, but there are a few implicit arguments going on in the background that need to be brought out.
First, the author mentions the benefit of transcending the sex and gender binaries (p. 3). But it’s unclear what the benefit is or why it is a moral benefit. What I mean is that by granting Halwani’s point that we don’t have control over sexual orientation, then the benefit from pansexuality wouldn’t be moral–or, if it is, how is it a moral benefit given that we don’t have control over it? I took the author to be indicating something like by raising greater visibility and changing political conditions to include pansexuality as a legitimate kind of sexual orientation, it hermeneutically benefits pansexuals because it helps them recognize and organize their experiences. But in order to get this benefit off the ground, the author would need to sharpen their criticism of the current projects they survey (pp. 8-12) because those don’t just fail on a conceptual or theoretical level, they also would cause the moral/hermeneutical harm of reclassifying pansexual experiences as something else under an inadequate theory. If this is the idea, then the author should come out and say so, or, if the author is aiming for another/different benefit from classifying pansexuality as a sexual orientation, then what is it? I think the author has a particular goal in mind, but it might help to define what they mean by “sex and gender justice” in order to bring the paper together.
Relatedly, without clarifying what the benefit is, and who receives the benefit, I worried that some of the phrasing was (unintentionally) smuggling in the idea that because pansexuality is more inclusive, it stands morally above the other kinds of sexual orientations.
A small point on p. 2, each of the celebrities the author names fall outside the demographic parameters from the Gallop Poll: MC was born in 1992, RS, was born in 1990, and BU was born in 1987, but Gallop defines Gen Z as people born between 1997-2003.
Another small point on p. 2: the author brings up the issue about about fantasies that do not need to be acted upon, but that are also not necessarily informative about our disposition. Although the author does revisit this point later on (p. 4), it wasn’t clear to me why sexual fantasies wouldn’t factor into the false identity claims here. So what if the fantasies aren’t acted up–wouldn’t they still show us something about the disposition? Or, since the “mistaken sexual identity” claim (p. 4) does show us that someone may fail to recognize an important fantasy as constitutive of a more accurate sexual identity, then why dismiss it as uninformative earlier (p. 2)?
Further, the dual goal in (1), of orientation and identity, needs a bit more work to show how they connect. The author approvingly cites Matt Andler’s distinction (p. 4) between a sexual orientation being a kind of psychological or personal sense of self, while the sexual identity imbues that orientation with a political meaning. Later on, (p. 10), the author touches on Dembroff’s ameliorative rationale for talking about sexual orientation in the first place–the reason we are interested it is to recast the legal landscape so that people who are gay, lesbian, or bi are not politically unable to marry, can be fired from their jobs, can exercise child custody rights, etc. But the only explicit political goal that the author cites is that Pansexual Pride Day increases visibility and awareness of pansexuality. Given that the author seems to be sharing Dembroff’s ameliorative rationale for classifying pansexuality as a sexual orientation (and identity a la Andler), this part should be a lot more developed. So, what is the political significance of publicly identifying oneself as pansexual? How does it affect employment? Levels of vulnerability to violence, homelessness, medical care discrimination? And, beyond a rights based account, how does it impact dating, sexual relationships, or other information interactions?
Minor point p. 4: the author distinguishes between sexual orientation and romantic orientation. I wasn’t sure I really understood what the distinction was supposed to accomplish. From the discussion here, I was left wondering why romantic attraction didn’t presuppose a sexual attraction–that is, I might want to date and woo someone, but if I don’t want to also have sex with them, then there seems to be a practical conflict with trying to woo. Or, would platonic dating be (and why not just call it hanging out?). But, even answering the questions, I wasn’t sure what the distinction was doing for the rest of the article–the author doesn’t go on to use it as a way to draw a more fine grained analysis about pansexuality different from other kinds of sexual orientations.
Phrasing concern, p. 4: Two clarifications on the bottom of page 4. First, the author seems to be trading on a stereotype about rural backwardness and urban progressiveness (see, for instance Mary L. Gray’s (2009, 2016) anthologies where this assumption isn’t as straightforward as we neatly talk about it in academia.
Second, the discussion might benefit by rephrasing the “passing” language when talking about trans women. The later discussion (p. 5) about gender presentation might be helpful to rephrase some of this point.
Phrasing concern, p. 5. The author seems to be referencing how dominant understandings of gender penalize people who are “discovered” to have genitals that do not “match” their gender presentation. Again, I take the author’s meaning, but I’d encourage them to try and rephrase this part of the paper–putting some distance between how the dominant understanding about gender establishes this requirement or something to help show that you’re just describing a less than idea situation rather than just stating facts.
On the whole, I found the discussion on why Vernallis’ account wouldn’t work for pansexuality quite persuasive. I did have one reservation, p. 7, where the author asserts that traits like vitality, charisma, charm, and grace are sources of attraction for pansexuals–that these “traits from within” enable or ground the attraction instead of bodily characteristics. But I wasn’t quite sure the overall claim in this paragraph lands, that these kinds of qualities (e.g., empathy, confidence, driven) transcend gender. The author does, to their credit, note that they are inflected by gender, but I was having a hard time coming up with a logical witness where these traits transcended gender. The closest I could come was the dominant stereotype in the 1980s and before, that gay men are shallow, sex-driven, etc. Seeing images of gay men caring for lovers, friends, neighbors, and strangers dying from AIDS related illnesses helped disrupt the stereotype because it showed an unmistakable compassion and care that we didn’t usually associate with men, but maybe the author could clarify what they mean by these traits transcending gender norms.
Circling back to the request for clarification about gender and sexual justice, the author uses an ethical objection to Stock’s arguments about intersex individuals (p. 9). I think the author’s responses to Stock work–that denying the number of cases makes them no less real; Stock is being intellectually lazy and dishonest here–but I wanted to see more about what sexual and gender justice looked like. The author rightly claims that we should recognize more than two sexes and that people who are intersexed should not be subjected to surgeries, but that seems much more a fact about the intersexual individuals and the add on about making the possible number of sexual orientation categories more accurate seemed to be ancillary (making it more about the people with the attraction rather than the people who are being harmed by the unnecessary surgical procedures). Spelling out what the ameliorative interest in pansexuality is clear earlier on might help then show why the moral concern for how we classify intersex individuals is, in fact, part and parcel of other kinds of sexual orientations.
I wasn’t clear (p. 10) if the author found Dembroff’s arbitrary distinction between druthers and sexual orientation to be a flaw for Dembroff or not. As I understood Dembroff, the difference between the two depends on the ameliorative rationale–that there isn’t any deep metaphysical reason about what makes a sexual desire a sexual orientation or a druther, but rather there are political consequences that go with some desires that make those orientations rather than druthers. Clarifying how you see this point factoring into your overall argument would help contextualize Dembroff’s contribution to your project as well as smooth the transition to your use of Halwani.
I found the presentation and exposition of Halwani very clear and intuitive, but the dialectic read a bit rushed. It wasn’t clear about why the well-being condition fails to apply to pansexuals (p. 11)--the author cites a previous discussion about applying BD to pansexuality, but why couldn’t Halwani’s argument be something like: pansexuality is a sexual orientation and being unable to act on it is a substantial loss of wellbeing, but these other traits (e.g., age, body type) enhance sexual experiences and amplify the good. By not having those amplifying features, pansexuals are missing out on that amplified good.
As above, the Halwani discussion concludes on the note that Pansexulaity might be a sexual identity, but it’s not clear to me what that means given that sexual identities (for Andler? The author?) have this deep political dimension.
In turn, I wasn’t sure how to read the last part of the paper, that pansexuality welcomes Halwani’s well-being condition given the above concerns the author cites.
On the whole, I think the paper hangs together a little loosely. Each section reads well, but I didn’t get a sense that there was an overarching argument driving through each section–not just to the next, but to the ultimate take-away. The author does point to this argumentative goal in terms of justice, but, as I mentioned above, I wanted to know what the nature of the benefit was (moral? Prudential? hermeneutical?), who benefited from it, and how did this (moral, prudential, hermeneutical) good connect with our current theories about sexual orientation so that there is a descriptive inaccuracy in addition to the above absence of moral, prudential, or hermeneutical goods.
Author Response
Dear Reviewer 3,
Please see my responses to the comments attached.
Thanks!

Round 2
Reviewer 1 Report
The structure and argument in the essay has been significantly improved, and is ready for publication, in my view. Well done to the author/s.
Author Response
Thank you!
Reviewer 3 Report
On the whole, the paper reads much smoother and purposeful. I wanted to thank the author for the revisions and believe that the paper is stronger. I only have one general, and small, worry, and then a few more specific points (listed below). I think the author successfully spells out the political goals of sex and gender justice, but while that part is clearer in the beginning, it seems to fade as the paper unfolds. What I mean is that while the author identifies cases where Pansexual Pride Day (p. 12) is important, they don't seem to say as much about the political or legal goals listed (p. 2) or overcoming the sex and gender binaries (p. 4). Now, a lot of benefits do come out en passant, but these also seem to be psychological, social, or hermeneutical, rather than issues of right-based opportunities (e.g., jobs, discrimination), resources (e.g., healthcare--are pansexuals denied healthcare? Given lesser care? Have to pay more for insurance?), or other general political involvement (e.g., if people who are pansexual are assaulted or disowned more). While the author does reference these points again in the conclusion, I wasn’t sure which “legal protections, the distribution of social goods such healthcare” the author had in mind. I don’t think this is a major worry, but it reads like the current version is still keeping some of the vestiges of the old here. I think the paper would go to the next level by either clarifying what and which, exactly, the benefits of recognizing pansexuality as a sexual orientation are—if they are political, what, specifically, are they? Are the benefits that they give newer demographics a more expansive vocabulary about sex and gender that positively changes our social imagination?
p. 1 phrasing: since the author numbers the sections, it might be helpful to keep that phrasing consistent throughout—the “the next section” bit through me because I wasn’t sure if the introduction was Section 1 or not.
p. 2 I really appreciated the explicit goals and listing the components of sex/gender justice. It gave me a much stronger sense of what the author had in mind.
Bottom of p. 2: you introduce the “…a trend that constitutes a sexual identity at best” as a way to (I presume) to foreshadow Matt Andler’s category (though you do reference it with Stock's work too). Just a small tweak/suggestion here: I think you could at add in that reference here to help orient readers who don’t know Andler’s work. So something like “…a trend that constitutes, at best, what Matt Andler has terms a “sexual identity” (XXXX). Or something to that effect to make it a bit more user-friendly and clarify what "sexual identity" means here.
p. 3 and 15: Thank you for going back and clarifying the age demographics for millennials and Gen Z in the main text and the footnote.
p. 15 Footnote 9. I think the clarification about sexual orientation itself being amoral is a good way to integrate why people are more ambivalent about Pope Francis than is usually depicted in media, though I’d encourage the author to reconsider the last sentence in the footnote: while I agree it does clarify the distinction, I have two worries: (1) as per the recent attacks on Stephen Kershnar, I worry that this might easily be misconstrued as the author endorsing pedophilia by saying there isn’t anything wrong with the inclination itself; (2) by textually linking the two sentences, about pedophilia coming immediately after the Pope Francis reference, it might be taken as a sideways swipe at Pope Francis/Catholicism. Perhaps the author is aware, intending, or indifferent to these two possibilities, but I thought it would at least be worthwhile to reflect on them—I leave it to the author to decide what they would want to do.
p. 4 The new revision about the moral implications of pansexuality reads a lot cleaner. I think the author is slightly underselling the worth of their project, though, because if the generational data shows that more younger people are identifying as pansexual, then the value also seems to be that it helps people who are pansexual recognize themselves as such, rather than thinking “I’m probably just bisexual” as older demographics might have done. So there seems to be a hermeneutical benefit too, though perhaps this points gets discussed later on.
I’m still quite sure I follow the response to the desire view. If a man has sexual fantasies about having sex with other men and doesn’t act on them, then I’m not sure why that doesn’t make him gay, bisexual, or pansexual—especially if the only reason is because there is political sanction (mutatis mutandis). The author elaborates that “for sexual desires towards a particular sex or gender are sometimes dismissed as mere fantasies that are not acted upon” but to rule out the desire view, the author would need to show that the dismissal is true. Or perhaps I’m misunderstanding what the example itself is? The behavioral objection, about sex in prison holds up well. Or, reading on, does the desire view in this paragraph differ in an important way from the attraction component that Diaz-Leon advocates for? The author revisits this point at the bottom of p. 5 while folding in Andler’s view, and again on the teleology issue on p. 7, but I’m still not clear on when the desire view would be insufficient.
p. 5 I appreciated revising the text to clarify the contrast between sexual and romantic orientation, especially in light of asexuality. Rephrasing the trans portion of the gender presentation worked really well too, thank you for revisiting it.
Small point about the writing: I think the pagination is still in the main text, though I think it needs to be at the end-note.
p. 7 The rewrite here is great. I liked that you turned to how pansexuals describe themselves in their own words to help clarify the line of argument.
Author Response
Dear Reviewer 3,
Thank you so much for your comments! Once again, they were extremely helpful, and I believe they made the paper even better.
I have added a paragraph to the last section, tying everything back to the sex and gender justice project described in Section 2. I’ve added your point about the hermeneutical benefit in the last sentence.
I’ve also moved the Andler reference to the paragraph you highlighted, when the concept of sexual identity is first introduced. I’ve also added another Andler reference to his 2022 chapter on the same topic since I find Andler’s work very intuitive and insightful.
Thank you so much for catching the wrong people I mentioned as examples of Generation Z! I am now very happy with all the folx I mention who identify as pansexual.
Regarding pedophilia, thank you for flagging this. Since the mention of pedophilia doesn’t actually add anything to my argument, I deleted the last sentence in the footnote.
Regarding homosexual fantasies in straight men, I’ve added, “It is said that…”. At that point in the paper, I am just beginning to introduce some key features of SO, as well as highlighting the commonsense views about it. The case of fantasies is under-described. Perhaps if it is a one-time thing, then it is not enough to render this person bisexual, as it is not a steady disposition to experience these. If it were, the answer would certainly be different.
I also fixed the page numbers for the Reference pages.
I am very grateful for the close attention you’ve paid to my arguments, and for the insightful comments that helped me make the paper even better! This is how peer reviewing should be done. Thank you.